# Single-cell RNA-sequencing data analysis reveals a highly correlated triphasic transcriptional response to SARS-CoV-2 infection

Pablo A. Gutiérrez[1,2 ✉] & Santiago F. Elena [2,3]

Single-cell RNA sequencing (scRNA-seq) is currently one of the most powerful techniques available to study the transcriptional response of thousands of cells to an external perturbation. Here, we perform a pseudotime analysis of SARS-CoV-2 infection using publicly available scRNA-seq data from human bronchial epithelial cells and colon and ileum organoids. Our results reveal that, for most genes, the transcriptional response to SARS-CoV-2 infection follows a non-linear pattern characterized by an initial and a final down-regulatory phase separated by an intermediate up-regulatory stage. A correlation analysis of transcriptional profiles suggests a common mechanism regulating the mRNA levels of most genes. Interestingly, genes encoded in the mitochondria or involved in translation exhibited distinct pseudotime profiles. To explain our results, we propose a simple model where nuclear export inhibition of nsp1-sensitive transcripts will be sufficient to explain the transcriptional shutdown of SARS-CoV-2 infected cells.

[1] Laboratorio de Microbiología Industrial, Facultad de Ciencias, Universidad Nacional de Colombia Sede Medellín, Carrera 65 Nro. 59A - 110, Medellín, Colombia. [2] Instituto de Biología Integrativa de Sistemas (I2SysBio), CSIC-Universitat de València, Paterna 46980 Valencia, Spain. [3] Santa Fe Institute, Santa Fe, NM 87501, USA. ✉email: paguties@unal.edu.co

Since the beginning of the SARS-CoV-2 pandemic in 2019, an unprecedented global research effort has taken place to elucidate the detailed molecular biology of this virus and its effects within infected cells[1–3]. Among the vast amount of information generated, single-cell RNA sequencing (scRNA-seq) datasets are particularly interesting as they can provide a detailed picture of the transcriptional state of individual cells at different stages during the infection cycle[4,5]. This is possible because the synthesis of cDNA libraries is performed on individualized cells and each cDNA product is labeled with an identifier that provides unique markers for each transcribed molecule[6–8]. Therefore, instead of describing the average transcriptional state of thousands to millions of cells, as in bulk RNA-seq, scRNA-seq captures a continuum of transcriptional states that can be used as a proxy to reconstruct the chronological response of a cell population to stimuli[9–11].

In virology, scRNA-seq was first used to study the heterogeneity of hepatitis C virus quasispecies in liver cells[12]. Since then, most virology scRNA-seq studies have focused on characterizing the heterogeneity of host cells infected with a wide range of viruses such as human papillomavirus, West Nile virus, yellow fever virus, Zika virus, or human immunodeficiency virus type 1, among others[5,13,14]. In addition, more recent studies have started to address questions related to the transcriptional dynamics of viruses and infected cells upon infection[5,15,16]. SARS-CoV-2 is not an exception, and since the emergence of the COVID-19 pandemic, dozens of papers have been published addressing different aspects of the molecular biology of this virus[17–20]. Unfortunately, despite the large volume of available data, most scRNA-seq investigations on SARS-CoV-2 have focused mostly on characterizing cell tropisms or molecular mechanisms at macroscopic timepoints along the infection cycle.

In this study, we present an analysis of the host cell response during the progression of SARS-CoV-2 infection using virus accumulation as a proxy of time. To obtain common features of the SARS-CoV-2 response, we performed a meta-analysis of three publicly available scRNA-seq datasets derived from in vitro inoculation studies of human bronchial epithelial cells[21], and colon and ileum organoids[22]. Our results revealed that the average transcriptional gene response to SARS-CoV-2 infection is non-linear and cannot be described adequately using conventional bulk RNA-seq analysis methods. For this reason, instead of conventional two-group comparisons (i.e., infected vs non-infected), our analyses involved pairwise comparisons of individual transcriptional profiles and the average transcriptional response to within-cell virus accumulation. Our results revealed that about 90% of genes exhibited the same qualitative response to infection, suggesting that the transcription of most genes is modulated by a global mechanism. Additionally, a different but highly correlated response was found among mitochondrial-encoded genes and genes involved in protein translation. A mechanism explaining the genome-wide transcriptional response to SARS-CoV-2 infection is proposed.

## Results

**The global transcriptional response to SARS-CoV-2 exhibits a wave-like pattern**. First, we tried to understand the global transcriptional response of infected cells by plotting the distribution of $\log_2$ fold changes ($\log_2 FC$) of transcripts as a function of the $\log_2$ of SARS-CoV-2 levels (Fig. 1a and Supplementary Data 1). This analysis indicated that the average transcriptional response displays an oscillatory behavior that can be divided qualitatively into three phases. The early phase occurs at low viral RNA levels ($\lesssim 1$ TPT) and is characterized by a global shutdown of transcription where 99.5% of genes in hBECs, 98.7% in colon, and 98.0% in ileum are downregulated. The intermediate phase occurs at medium viral loads and is characterized by an increase in expression levels with maximum fold changes between 1.4 and 4.1. Finally, the late phase corresponds to the highest SARS-CoV-2 levels and was characterized by a second downregulation of cellular transcripts.

As the oscillatory transcriptional response to SARS-CoV-2 might bias conventional bulk differential gene expression (DGE) analyses, we tested the effect of selecting cells from different phases as contrasting groups. In Fig. 1b, we show the volcano plots that result from comparison with all the infected cells, in addition to comparison involving early, intermediate, and late phase cells (Fig. 1b). These results unveil that the classification of genes as down- or up-regulated is highly dependent on how the contrasting cell populations are selected. For example, in hBECs, the bulk DGEs analysis revealed 9429 down- and 481 up-regulated genes; and similar results were obtained using cells from the early phase (9710 down- vs 200 up-regulated genes). However, this trend is reversed when cells from the middle phase are used: 263 down- vs 9647 up-regulated genes are now observed. Finally, as expected from the distribution of $\log_2 FC$ values, DGE trends are reversed again when cells from the late phase are compared (9,810 down- vs 99 up-regulated genes). Similar results were obtained in the analysis of infected colon and ileum cells (Fig. 1b and Supplementary Data 3).

To better compare the global transcriptional response at different stages of the viral infection cycle, we plotted the average scaled gene expression values at different viral loads (Fig. 1c). To do this, expression values at each $\log_2$ interval were transformed into $z$-scores with respect to the mean and standard deviations of uninfected cells. This correction removes biases in the mean response patterns that might result from highly-expressed genes. After this transformation, intervals of up- or down-regulation correspond to positive and negative values, respectively. Data ordered in this way can be interpreted as a pseudotime, which, here, represents doublings in viral concentration[9,11,23]. In hBECs, $z$-scores fluctuated between −1.2 and 0.9, and transitions between phases occurred at pseudotimes of 3.5 (~10.3 TPT) and 6.2 (~72.5 TPT) (Fig. 1c). In colon- and ileum-organoids downregulation during the early phase was less pronounced but exhibited a stronger upregulatory response in the intermediate phase. In these cases, $z$-scores fluctuated between −0.5 and 2.3 in the colon, and −0.3 and 4.7 in the ileum. Transition points occurred at pseudotimes of 4.9 (~28.9 TPT) and 11.4 (~2,701.3 TPT) in the colon, and 4.0 (~15 TPT) and 12.7 (~6653 TPT) in the ileum (Fig. 1c). To verify the specificity of the SARS-CoV-2 response, we performed a similar analysis on ileum cells infected HAstV1 (Supplementary Fig. 3) which revealed a different profile characterized by a rapid drop in transcription levels followed by a steady increase at high virus accumulations. A strong wave-like behavior was not observed in this case.

**Classification of gene responses**. Next, we proceeded to identify genes with abnormal transcriptional profiles using two metrics: the root-mean-square deviation (RMSD) to the average response vector and the magnitude of the initial response ($\Delta_0$) (Fig. 2a). The latter was used to give a stronger weight to datapoints at earlier infection times and to select for genes where the initial transcriptional response was either weaker or stronger than expected. A transcriptional profile was considered an outlier when either of these parameters was outside the intervals defined by the 1.5×IQR rule. Our results indicate that most cellular transcripts have a response pattern qualitatively similar to the mean response vector, with only 7.2 to 10.1% of genes exhibiting atypical transcriptional profiles (Fig. 2b and Supplementary

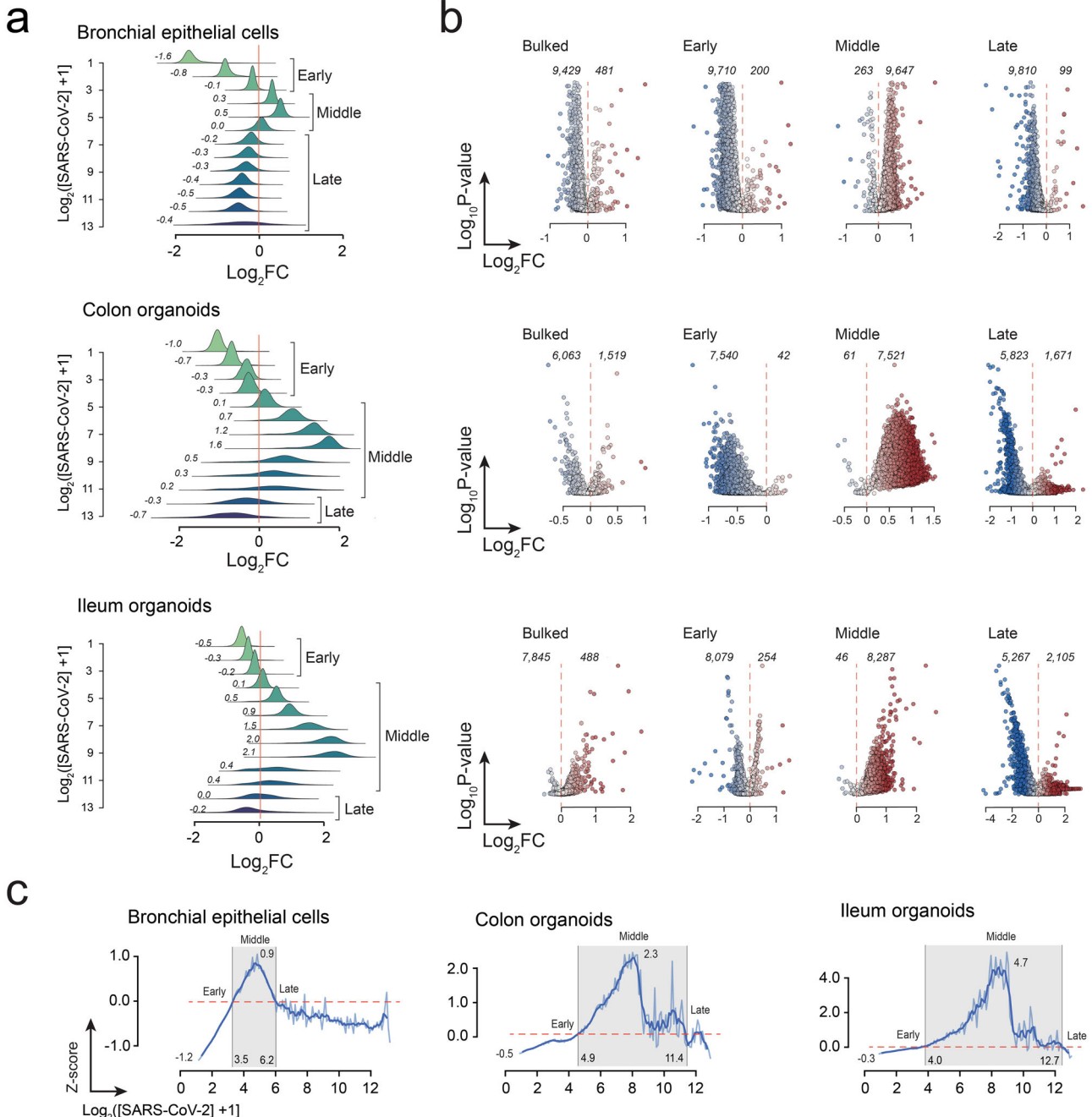

**Fig. 1 Global transcriptional response to SARS-CoV-2 in human bronchial epithelial cells, colon- and ileum-organoids. a** Distribution of log$_2$FC values for all genes plotted against SARS-CoV-2 reads using a log$_2$ scale. Mean log$_2$FC values at each interval are shown to the left of each histogram. **b** Volcano plot analysis of infected cells with respect to noninfected cells using bulked cells, and subgroups comprising cells at early, middle, and intermediate phases of infection. For clarity, corrected *P* values were removed from the ordinate axis. **c** Mean transcriptional response of each cell type. Each plot was calculated from individual gene profiles scaled using *z*-scores with respect to the transcript levels observed in uninfected cells. These plots confirm the global oscillatory transcriptional profile of genes at increasing SARS-CoV-2 levels.

Data 3); 183 outliers were shared between the three cell types (Fig. 2c). A gene ontology (GO) analysis of all the outliers from the three cell types (Fig. 2d) revealed that this subset was enriched in genes involved in immune response (e.g., GO:0030593, GO:0031640, and GO:0061844), translation (e.g., GO:0042274, GO:0000028, GO:000641, and GO:0002181), respiration (e.g., GO:0006120 and GO:0042776), and control of cellular proliferation and death (e.g., GO:0001895). The subset was also significantly depleted in genes involved in intracellular protein transport (e.g., GO:0006886), protein phosphorylation (e.g.,

GO:0006468), and regulation of transcription (e.g., GO:0006355 and GO:0006357).

As the differential transcriptional response might be the result of general regulatory mechanisms induced or inhibited by SARS-CoV-2, we decided to interrogate the data to identify common responses involving interferon signaling, transcription factors, and RNA binding proteins.

**Transcriptional response of IFN-stimulated genes**. At early infection times, SARS-CoV-2 downregulates the expression of

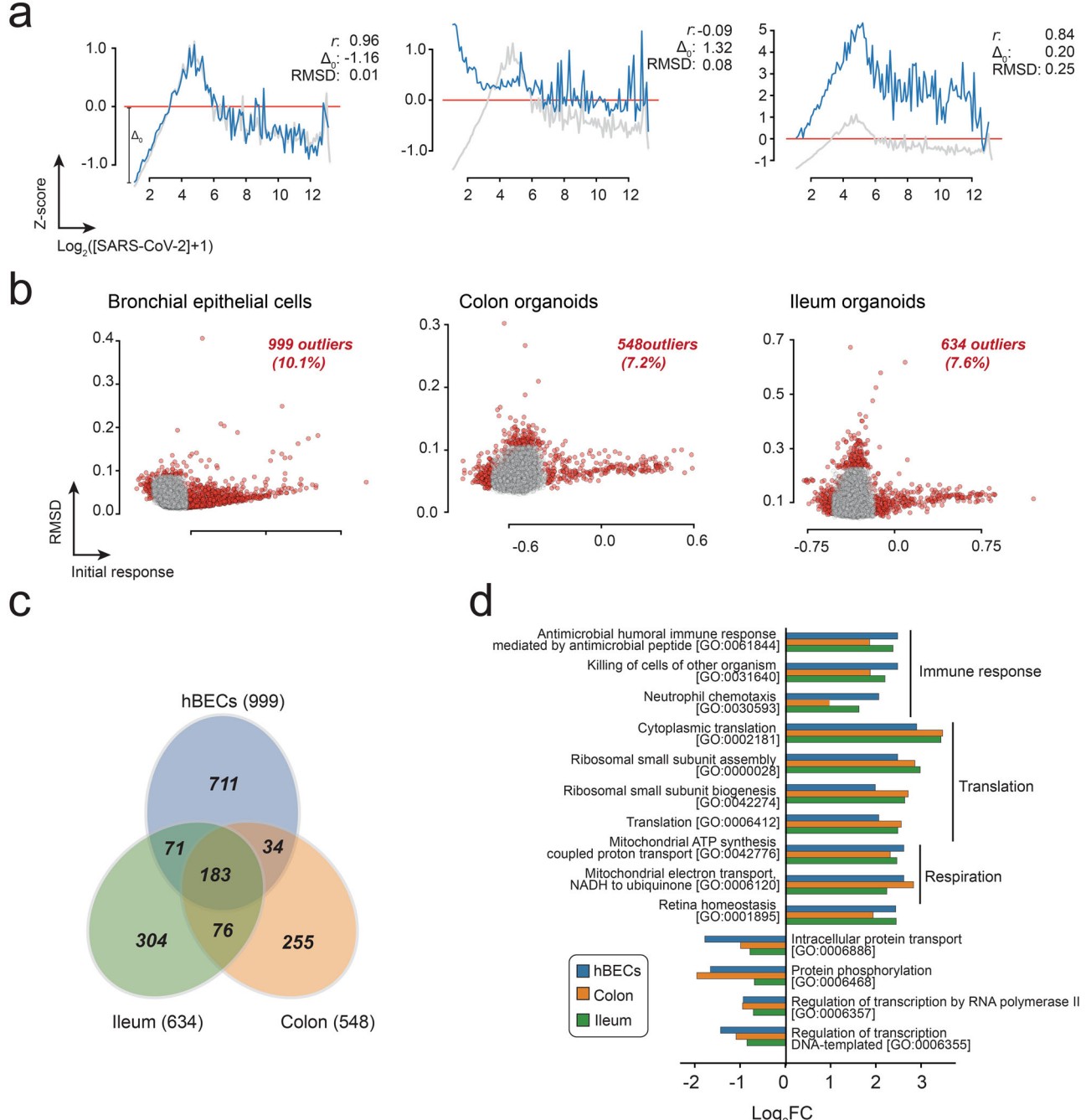

**Fig. 2 Classification of genes based on their similarity to the mean response vector. a** Examples of selected individual transcriptional profiles. Individual transcriptional responses (blue) were compared to the mean response (gray) using the root-mean-square deviation (RMSD) and the magnitude of the initial response ($\Delta_0$). The Pearson correlation ($r$) for each comparison is also provided. **b** A plot of RMSD and initial response values for all genes, allows the identification of genes exhibiting abnormal transcriptional responses (outliers, red). **c** Venn diagrams illustrating the distribution of outliers among the three datasets used in the analysis. **d** GO-enrichment analysis of outliers from the three datasets indicating that most outliers correspond to genes involved in immune response, translation, and cellular respiration.

IFN-stimulated genes (ISGs) involved in antiviral defense[24–27]. We then decided to investigate whether genes involved in the IFN response follow an abnormal transcription pattern upon SARS-CoV-2 infection. Firstly, we analyzed the transcriptional profiles of genes encoding for MDA-5 and RIG-I, which are the main intracellular sensors that recognize double-stranded RNAs and trigger the innate immune response upon infection[27–29]. Analysis of the transcriptional profiles of MDA-5 and RIG-I, encoded by genes *IFIH1* and *DDX58*, respectively, revealed that both have a transcriptional response like the average response vector (Fig. 3a

upper panels and Supplementary Data 3). This result suggests that the transcriptional shutdown of the IFN response induced by MDA-5 and RIG-I is probably not specific, and results from a more general mechanism affecting most genes in the infected cell.

We then proceeded to analyze the response of individual ISGs in the three datasets. To do this, we selected genes using a recently curated and validated dataset of ISGs[30]. A total of 238, 179, and 219 ISGs were identified in hBECs, colon, and ileum, respectively (Supplementary Data 3). As expected, a GO analysis revealed that most of the identified ISGs are involved in the defense response to

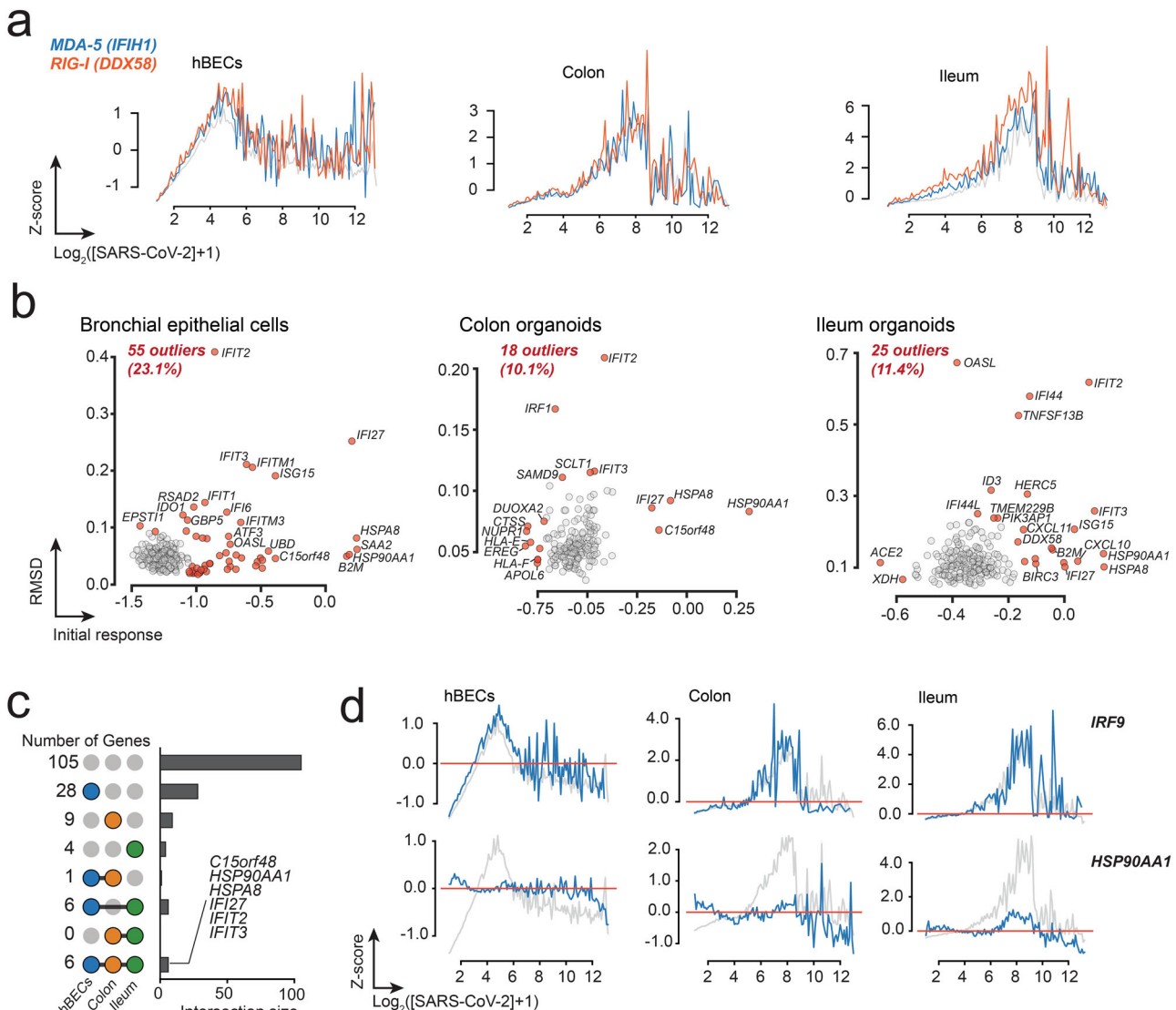

**Fig. 3 Global transcriptional response of ISGs in human bronchial epithelial cells, colon-, and ileum-organoids. a** The transcriptional response of the cytoplasmic viral receptors MDA-4 and RIG-I correlated with the mean transcriptional response. Enrichment of GO terms associated with both signaling pathways are indicated. **b** Distribution of *RMSD* and $\Delta_0$ profiles of ISGs. Genes considered to be outliers are shown in red. **c** Upset plots illustrating the number of outlier ISGs shared among the three datasets. The first row corresponds to ISGs exhibiting an average response in the three datasets. The six genes exhibiting a differential response to SARS-CoV-2 infections in the three cell types are indicated. **d** Representative examples of ISGs exhibiting average (*IRF9*) and abnormal (*HSP90AA1*) transcriptional responses to SARS-CoV-2 infection.

viruses (Supplementary Fig. 3). Like the MDA-5 and RIG-I receptors, 77–90% of ISGs exhibited an average transcriptional response to SARS-CoV-2 indicating that the mRNA levels of these genes are negatively regulated at the onset of infection and their response is probably a consequence of the global transcriptional shutdown (Fig. 3b). Examples of ISGs exhibiting different response are shown in Fig. 3d. An analysis of abnormal transcriptional responses identified 55 genes from hBECs, 18 from colon and 25 from ileum; six of them common to the three datasets: heat shock protein HSP90α (*HSP90AA1*), interferon-induced protein with tetratricopeptide repeats 2 (*IFIT2*), interferon-induced protein with tetratricopeptide repeats 3 (*IFIT3*), heat shock cognate 71 kDa protein (*HSPA8*), normal mucosa of esophagus-specific gene 1 protein (*C15orf48*), and interferon alpha-inducible protein 27 (*IFI27*) (Fig. 3c). HSP90AA1 and HSPA8 are chaperone proteins which are typically required for viral replication[31]. In addition, HSPA8 has been shown to be an attachment factor for avian infectious

bronchitis coronavirus[32]. *C15orf48* expression, on the other hand, has been shown to be involved in the mitochondrial stress response (MISTR) and is expressed by pathogenic macrophages in severe COVID-19[33,34]. Interestingly, IFI27 and IFIT3 are also involved with the mitochondrial processes; the former locates in the nuclear inner membrane and is indispensable for mitochondrial function[35], and the latter is a mediator of the mitochondrial antiviral signaling (MAVS) complex triggered by the MDA-5 and RIG-I signaling pathways[36]. However, a connection between these genes and the global transcriptional profiles observed upon SARS-CoV-2 infection does not seem obvious.

**Transcriptional response of transcription factors.** To further understand the effect of SARS-CoV-2 on transcription factors (TF), which could explain the initial transcriptional shutdown, we investigated the effect of SARS-CoV-2 on individual transcription factors from a curated collection of known and likely human transcription factors[37]. Our analysis identified a total of 785, 448,

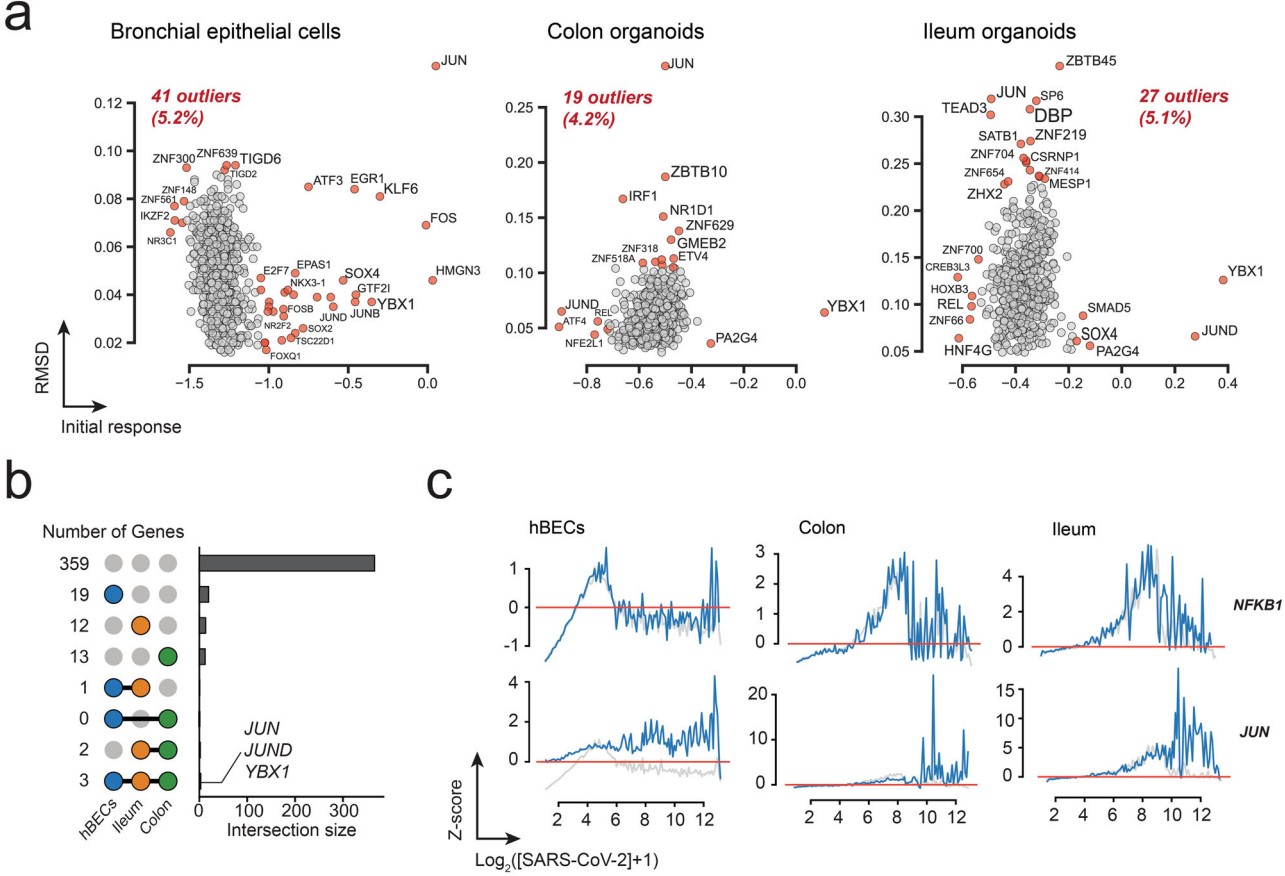

**Fig. 4 Global transcriptional response transcription factors. a** Distribution of transcriptional responses of identified transcription factors from hBECs, colon, and ileum datasets. Outliers are colored red. **b** Upset plots illustrating the number of outlier IFs shared among the three datasets. Most TFs (359) were common to the three datasets and exhibited an average response. The three TFs exhibiting a differential response to SARS-CoV-2 infections in the three cell types are indicated. **c** Representative examples of TFs exhibiting average (*NFKB1*) and abnormal (*JUN*) transcriptional responses.

and 531 TFs in hBEC, colon, and ileum, respectively, with 409 TFs in common to the three datasets (Supplementary Data 3). Again, most transcriptions factors behaved like the mean response vector, and only 36 TFs in hBECs, 16 in colon, and 22 in ileum were classified as outliers (Fig. 4a and Supplementary Data 3). Three TFs were found to be differentially regulated in the three datasets: the Y-box binding protein 1 (*YBX1*), and the proto-oncogene TFs *JUN* and *JUND* (Fig. 4b and Supplementary Data 3). *YBX1* encodes a highly conserved cold shock domain protein with DNA and RNA binding properties implicated in the regulation of transcription and translation, pre-mRNA splicing, DNA reparation, mRNA packaging, cell proliferation, stress response, and apoptosis. It has been shown to be a component of messenger ribonucleoprotein (mRNP) complexes with a role in microRNA processing[38,39]. The product of gene *JUN* is a component of the AP1 TF complex involved in oncogenic transformation[40] and is targeted to the nucleolus, where it seems to modulate nucleolar architecture and ribosomal RNA accumulation[41]. The expression profile of *JUN* stands out as its transcriptional profile suggests transcriptional activation at later infection times and its similar response in the three datasets suggests that this gene might be a marker that induces apoptosis in infected cells (Fig. 4c). *JUND* encodes a JUND proto-oncogene that is a functional component of the AP1 TF complex and has been proposed to protect cells from p53-dependent senescence and apoptosis[42]. These results suggest that the transcriptional response of most TFs is non-specific; however, exceptions

probably include TFs expressed after the initial transcriptional shutdown induced after the activation of cell-death pathways at late SARS-CoV-2 infection stages. Again, a connection between *JUN, JUND,* or *YBX1* and the global transcriptional response is unclear.

**RNA binding proteins.** As levels of mRNA might be modulated by their interaction with RNA binding proteins (RBPs), we explored the transcriptional profiles of genes from a dataset of canonical RBPs from the eukaryotic RBP database, EuRBPDB[43]. More than a thousand RBPs were identified in each dataset (Fig. 5a and Supplementary Data 3), of which ~10% exhibited abnormal transcriptional profiles (Fig. 5b). Interestingly, 83 abnormal responding genes were shared between the three datasets (Fig. 5b) which contrast with the analysis of ISGs and TFs where only a handful of abnormal responding genes shared among datasets. Figure 5c shows illustrative examples of representative RBPs with average (*SRPNP35*) and abnormal (*RPS5*) transcriptional responses. GO analysis of genes abnormal responding genes revealed that these were enriched in terms involved in the synthesis of proteins involved in translation (Fig. 5d, lower panel). A detailed analysis of the outliers revealed that the common response involved 41 large ribosomal subunit proteins, 30 small ribosomal subunit proteins, four elongation factors (*EEF1A1, EEF1B2, EEF1G,* and *EEF2*), the FAU ubiquitin-like and ribosomal protein S30 fusion (*FAU*), the histidine triad nucleotide-binding protein 1 (*HINT1*), the heterogeneous nuclear

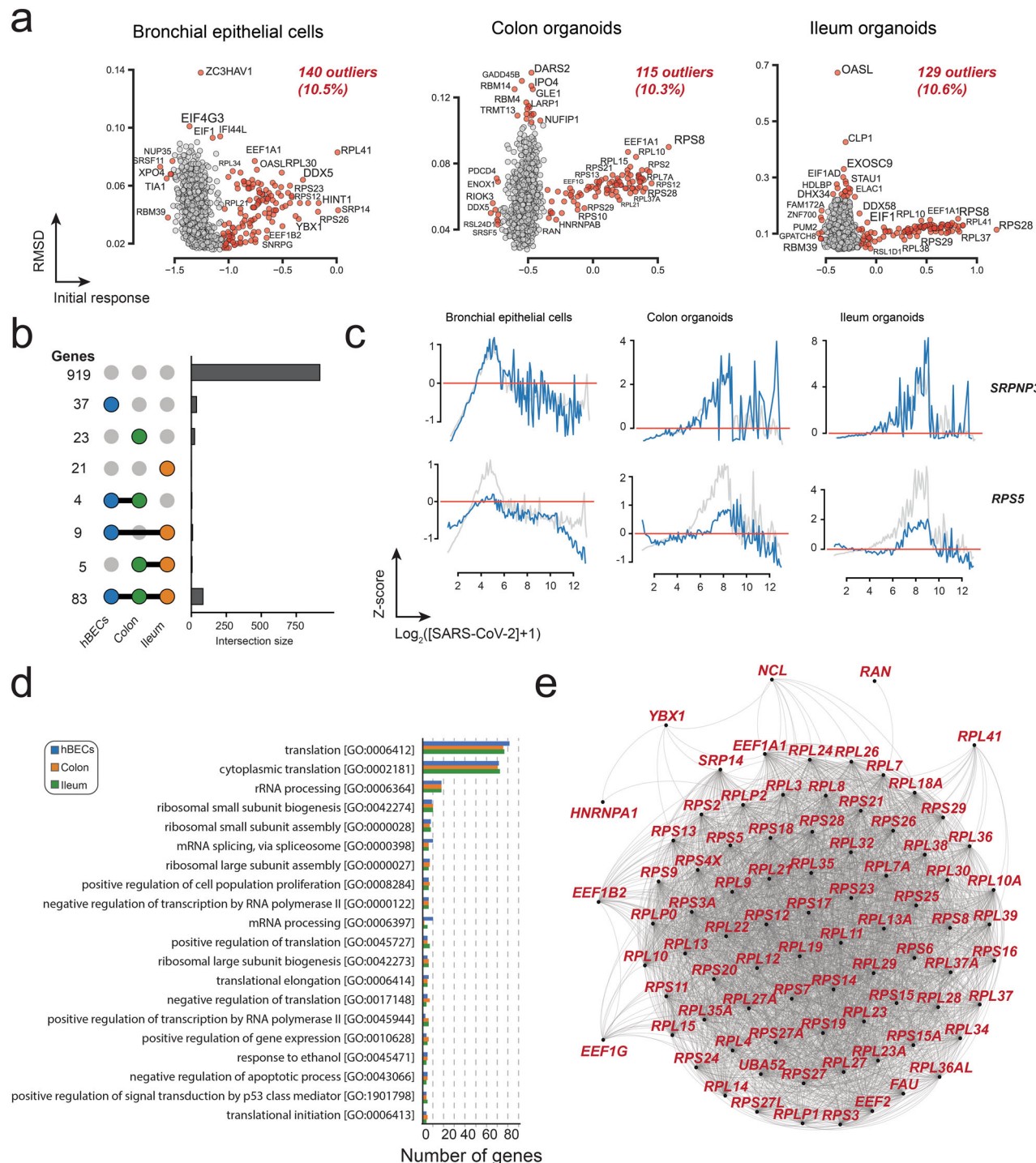

**Fig. 5 Transcriptional response of RNA binding proteins. a** Distribution of transcriptional responses of RBPs identified in hBECs, colon, and ileum datasets. Outliers are colored in red. **b** Upset plots illustrating the number of outlier RBPs shared among the three datasets. In contrast to ISGs and TFs, a large number of RBP outliers were common to the three datasets (83). **c** Representative examples of RBPs with average (*SRPNP35*) and abnormal (*RPS5*) transcriptional responses. **d** GO annotation of the 25 most common terms of RBPs with normal and abnormal transcriptional responses. **e** A STRING database search of differentially regulated RBPs revealed a highly connected functional network suggesting that transcripts involved in translation are regulated differently with respect to most genes.

ribonucleoprotein A1 (*HNRNPA1*) involved in pre-mRNA processing in the nucleus, nucleolin (*NCL*) involved in ribosome processing, the ras-related nuclear protein (*RAN*) essential for the translocation of RNA and proteins through the nuclear pore complex, the signal recognition particle 14 (*SRP14*) involved in

protein targeting to ER, and the ubiquitin A-52 residue ribosomal protein fusion product 1 (*UBA52*) involved in protein degradation by the 26S proteosome, and the *YBX1* gene identified in the previous TF analysis. Finally, a search in the STRING database, revealed that most abnormal responding RBPs are part of a highly

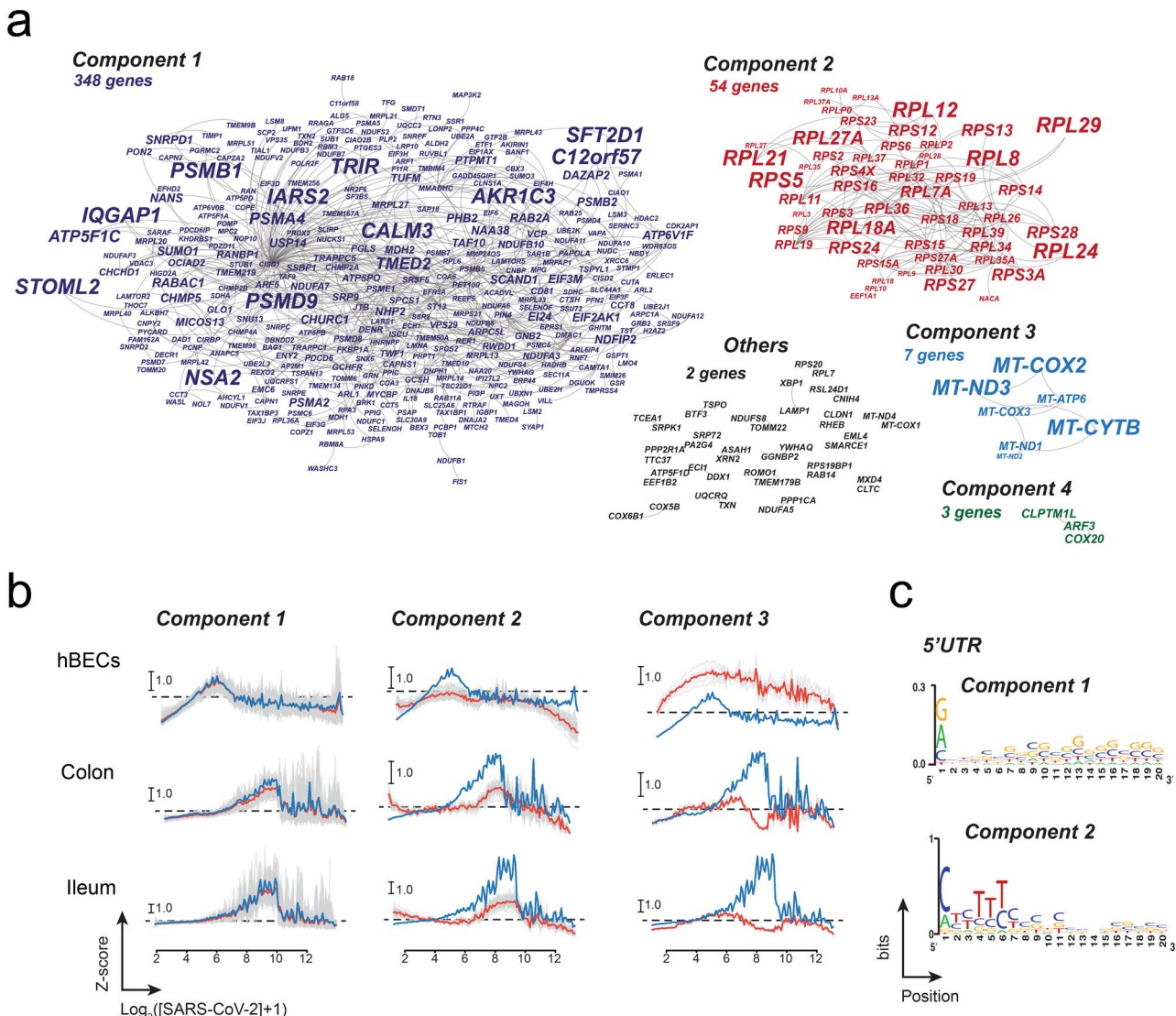

**Fig. 6 Correlated responses to SARS-CoV-2 infection common to the three cell types. a** The network was constructed by connected pairs of genes exhibiting a significant correlation in their transcriptional profiles in the three datasets. This analysis resulted in a network with 25 connected components comprising between 348 to 2 genes. Node size is scaled to represent the clustering coefficient. **b** The transcriptional profiles of the three largest connected components revealed distinct response profiles associated with the mean response (component 1), genes involved in translation (component 2), and mitochondria-encoded genes (component 3). In each example, profiles of individual genes are colored gray; the average transcriptional response of genes within each module is shown in red. For comparison, the mean response profile is shown in blue. **c** Sequence logos of the 5′ UTRs reveal the presence of characteristic sequence signatures for genes belonging to modules 1 and 2.

connected physical network, which suggests a distinct response to SARS-CoV-2 infection of genes implicated in translation (Fig. 5e).

**Gene regulatory networks**. As gene expression is often modular and usually involves the coordinated expression of functionally related genes[9], we investigated the correlation of transcriptional responses between pairs of genes in the three datasets. First, we constructed individual correlation networks using the normalized transcriptional profiles (Supplementary Data 2, 4). Then, we used the individual networks to construct a general response network comprising all the edges shared between the three individual networks (Fig. 6a). The resulting network comprised 25 connected components, the largest of which consisted of 348 genes that exhibited transcriptional profiles similar to the mean response vector (Fig. 6b). The second largest connected component comprised 54 genes, and included mostly ribosomal protein

genes, and proteins involved in translation. In this case, all genes within the network exhibited a qualitative behavior that was characterized by weaker downregulation and upregulation of transcription during the early and middle phases of infection as compared to the mean response vector (Fig. 6b). Interestingly, the third largest connected component was comprised exclusively of mitochondrially encoded genes, suggesting that response of mitochondrial genes upon SARS-CoV-2 infection follows a different mechanism from the global response and genes involved in translation (Fig. 6b). The remaining connected components comprised between two to three genes and some of them exhibited transcriptional responses like those observed in the largest components (Supplementary Data 4); they were no included as part of the largest subnetworks as the some of the edges failed to meet the filtering criteria. Previous studies have suggested a role of the 5′ UTR of mRNAs in the ability to escape the global suppression of translation induced by SARS-CoV-2

and related coronaviruses[44–46]. To investigate the presence of potential sequence motifs in transcripts from the largest connected components, we generated a sequence logo using the first 20 nucleotides of reference sequences from these clusters. This analysis suggests that transcripts from the first connected component tend to have A, C, or G as the first nucleotide with equal probability and higher GC content between positions 7–20 (Fig. 6c). In contrast, sequences from the second connected component contain TOP-motifs in their 5′ UTRs, which is to be expected for transcripts encoding ribosomal proteins and components of the translation apparatus[46,47]. Interestingly, analysis of protein translation in cells expressing SARS-CoV-2 protein Nsp1 revealed that the translation efficiency was slightly higher for genes with lower GC content in their 5′ UTR[46]. Finally, an analysis of the transcriptional profiles in cells infected with HAstV1 did not reveal a distinct correlated response in genes from modules 1, 2, and 3 (Supplementary Fig. 3), suggesting that transcriptional clusters are probably specific to SARS-CoV-2.

## Discussion

Our analysis indicates that the global transcriptional response to SARS-CoV-2 infection exhibits a behavior characterized by three phases: an initial downregulatory phase, a middle upregulatory phase, and a late downregulatory phase. Clearly, this type of response cannot be analyzed using the standard two-group comparisons typically used in bulk RNA-seq methods (e.g., volcano plots)[48–50]. This is of special concern in the case of viral infection studies, as macroscopic timepoints (e.g., days or hours) comprise a complex mixture of cellular states that must be disentangled prior to running any biologically meaningful analysis. Failure to do this can introduce biases in the selection of DGEs as opposing regulatory trends in cells at different transcriptional stages might cancel each other. Analysis of the viral infection process should probably include other metrics, and as suggested here, transcriptional profiles might be a better representation of the response of individual genes along the infection timeline. Standard filtering procedures of cells in scRNA-seq are also another important source of bias, as is customary to discard cells with a low number of mapped genes and/or a high proportion of mitochondrial transcripts[51–54]. This filtering strategy might not be adequate in studies of viral dynamics as, at late infection times, cells are expected to be dominated by viral transcripts and the diversity of transcribed genes is expected to be low, thus removing cells that might provide valuable information on the molecular events happening in the late stages of viral infection. In our opinion, in single-cell virology studies, cells should be selected with respect to the number of individual transcripts and not the total number of mapped genes. A similar logic applies to the removal of cells with high levels of mitochondrial-encoded transcripts, as overexpression of mitochondrial genes might represent a legitimate response to viral infection, as has been shown for SARS-CoV-2[55–57].

Considering our observations, we wish to propose an updated model of SARS-CoV-2 infection. A good model must be able to explain: (i) the transcriptional downregulatory phase at early infection times exhibited by about ~90% of genes, (ii) the selective transcriptional modulation of mitochondrial-encoded genes and genes involved translation, (iii) the global increase in host transcript levels observed at the middle of the infection cycle, and (iv) the second downregulatory phase at late infection times. This behavior seems to be a feature of SARS-CoV-2 infection as a similar analysis performed on ileum cells infected with HAstV1 did not reveal a correlated response for most genes, and did not exhibit an intermediate upregulatory phase (Supplementary Fig. 3 and Supplementary Data 6).

We believe that the multiprong strategy proposed by Finkel et al.[26], in combination with the known biology of the viral nonstructural protein 1 (nsp1), can explain the results presented here. The multiprong strategy explains the shutdown of host protein synthesis induced by SARS-CoV-2 as a combination of three effects: a global inhibition of protein translation, degradation of cytosolic cellular transcripts, and blockage of nuclear mRNA export[26]. Additionally, it is also well-established that nsp1 is responsible for orchestrating the transcriptional and translational shutdown induced by SARS-CoV-2. First, nsp1 is a strong inhibitor of translation, affecting translation of both host and viral mRNA[58] by docking its C-terminal domain to the mRNA entry channel of the 40S ribosomal subunit[25,59,60]. Second, the expression of nsp1 is sufficient to induce the global degradation of host mRNAs[26,44,61–63] by an unknown mechanism independent of a viral encoded RNase and/or ribonuclease L[62,64]. Third, immunoprecipitation and mass spectrometry studies have shown that the N-terminal domain of nsp1 interacts with the mRNA export receptor protein NFX1 preventing its binding to mRNA export adapters that results in the accumulation of mRNA in the nucleus[65]. Clearly, nsp1 is the key factor involved in shutting down host protein expression.

The observed downregulatory phase is well-supported by many studies that have shown that soon after infection, SARS-CoV-2 mRNA transport is stalled and there is a rapid degradation of cytoplasmic mRNA[26,44,61–63]. Some believe that the inhibition of nuclear mRNA export is a direct consequence of the widespread mRNA degradation in the cytosol[62]. However, we believe the opposite to be true and propose a mechanism where the downregulatory phase results from altering the natural steady-state dynamics between mRNA degradation and export to the cytoplasm (Fig. 7). The levels of mRNAs in the cytoplasm are determined by a steady-state equilibrium between the rates of transcription, nuclear export, and natural RNA turnover in the cytoplasm[66], therefore, a global blockage of mRNA export will compromise the input of newly synthesized mRNA into the cytoplasm resulting in global decrease of cytoplasmic mRNA levels and an increase in concentration in the nucleus (Fig. 7b). In other words, expression of SARS-CoV-2 nsp1 leads to a cellular state where mRNA is stalled at the nucleus and cannot replace the mRNA being degraded in the cytoplasm by its natural turnover rate (Fig. 7b). This is supported by RNA-seq data on infected cells that revealed nuclear accumulation of mRNA and a reduction of mRNAs in the cytoplasm and increased levels of intronic reads[26,67]. Moreover, the mRNA export block precedes the reduction in mRNA levels and only requires expression of nsp1;[65] and a study of IFNB1 induction upon SARS-CoV-2 infection found that soon after infection, transcribed mRNAs fail to disseminate from transcriptional foci and are preferentially retained in the nucleus[62].

Inhibition of nuclear transport by nsp1 can also explain the differential transcriptional response of transcripts involved in translation and mitochondrial-encoded transcripts. Most nuclear-encoded transcripts are exported into the cytoplasm by a mechanism involving the heterodimeric export receptor NXF1·NXT1[68] and it has been shown that nsp1 displaces NXF1 from the nuclear pore complex, impairing the docking of mRNA[65]. However, some transcripts can be exported by alternative pathways, indeed, regulation of mRNA export is a mechanism used to modulate several critical biological processes such as DNA repair, stress response, and maintenance of pluripotency[68]. We postulate that host transcripts can be classified into nsp1-sensitive and nps1-insensitive. In our view, nsp1-sensitive transcripts correspond to transcriptionally active loci at the time of infection that are blocked by the interaction of nsp1 with NXF1. These transcripts correspond to about 90% of the

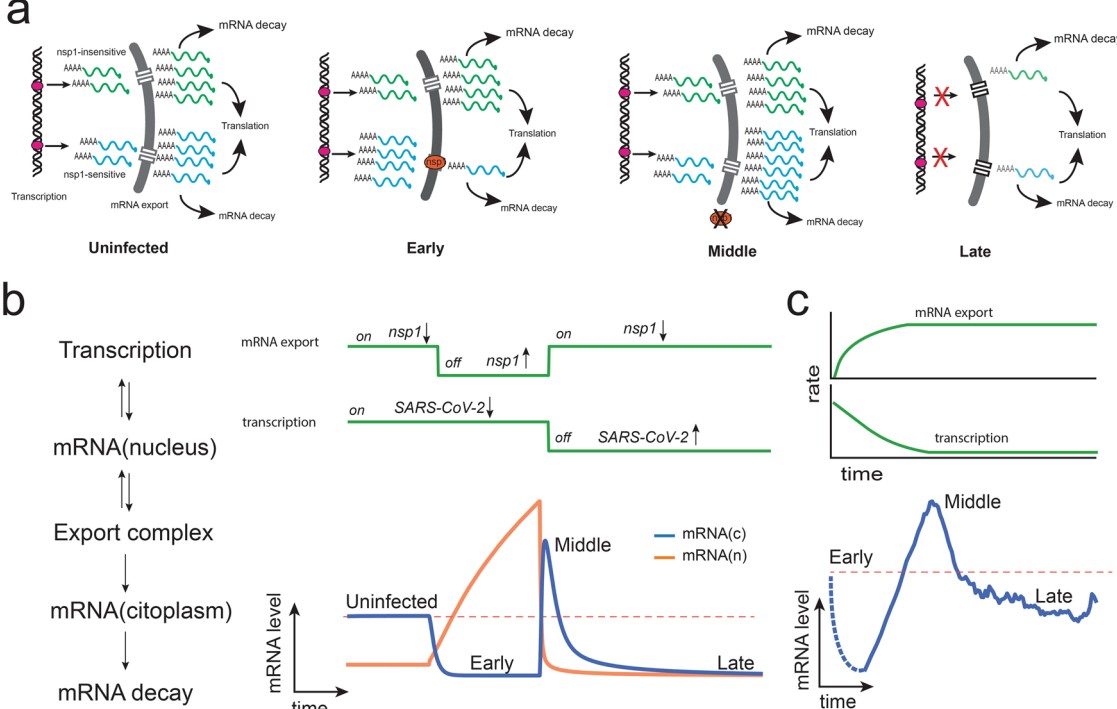

**Fig. 7 Proposed model to explain the transcriptional response of infected cells to SARS-CoV-2 infection. a** Selective nuclear export of nsp1-sensitive transcripts will result in the degradation of cytoplasmic mRNAs and their accumulation in the nucleus. Once the blockage is removed, retained transcripts are released, resulting in a rapid increase in cytoplasmic mRNA levels. A shutdown of transcription during the late infection stage will result in a global decrease of host mRNAs. **b** A simplified dynamical model simulating nuclear mRNA export and transcription as on/off switches is enough to model the mean transcriptional profile. **c** Reinterpretation of the mean transcriptional profile of hBECs. In contrast to the on-off switch model presented in panel **b**, it is most likely that the shutdown of transcription and nuclear export occurs gradually, as shown in the upper panel. The dotted lines represent early infection states that could not be detected in the present study.

transcriptome, including genes involved in the interferon response. This could explain the differential response of mRNAs involved in translation. Interestingly, a selective nuclear export mechanism of mRNAs related to translational function, including ribosomal proteins, have been shown to exist involving protein TDP-43[69], an RNA binding protein involved in the regulation of transport and translation of mature mRNA in the cytoplasm[70]. The differential regulation of genes involved in translation has been independently demonstrated by several studies, which revealed that genes with 5′ terminal oligopyrimidine tracts tend to escape the suppression of translation induced by SARS-CoV-2[46] and replacement of these sequences with purines results in reduced translational efficiency[46]. However, we believe that possessing the TOP motif is not a sufficient condition for the selective transport of translational mRNAs, as several TOP containing sequences are part of the mean response subset (Supplementary Data 3). Selective mRNA export inhibition also explains the differential response of mitochondrial-encoded genes as these genes do not require a nuclear export mechanism and should not be affected by the nsp1 blockage. The differential regulation of mitochondrial-encoded genes is also well-supported by experimental evidence that showed that they are less affected by SARS-CoV-2 infection than cellular transcripts[26,46,56].

The upregulatory response observed during the middle phase of SARS-CoV-2 infection is, to the best of our knowledge, an unexpected observation that deserves additional discussion. If we assume that nsp1 is degraded or inactivated after it has accomplished its function, then the stalled mRNA transcripts can be finally exported to the nucleus in a rapid burst which would look like an upregulatory event during the intermediate infection stage (Fig. 7). This rapid burst of newly synthesized transcripts can also

explain the late but strong response of the interferon system typical of COVID-19 disease[63,71,72]. It is also reasonable to assume that, when the nuclear blockage stops, it is already too late for the host cell to outcompete the replicating virus, which results in the final downregulatory phase observed at high SARS-CoV-2 levels. Qualitatively, this behavior can be simulated with a simple dynamical model where global mRNA export and transcription act like switches (Fig. 7a). Expression of *nsp1* acts like an off-switch for nsp1-sensitive transcripts that results in their accumulation in the nucleus while nsp1-insensitive transcripts can transit normally to the cytoplasm. The middle phase corresponds to the inactivation of the nsp1-induced blockage, which releases the stalled transcript to the cytoplasm that results in the increased transcription levels observed during the middle phase. Finally, the late phase corresponds to a shutdown of transcription that also results in a global decrease in transcript levels.

In summary, our analysis reveals a complex transcriptional response to SARS-CoV-2 infection that can be easily explained using the blockage of mRNA transport as the key molecular event. We believe that this model provides a parsimonious explanation of the host transcriptional and translational shutdown induced by SARS-CoV-2 and suggests that targeting the nsp1 ability to disrupt nuclear export might be the key to counteract the effect of this virus within infected cells.

## Methods
**Data**. Raw sequencing files were downloaded from the Sequence Read Archive (SRA) at NCBI using the SRA toolkit 2.11.1 (https://www.ncbi.nlm.nih.gov/sra/) and included previously published scRNA-seq data on SARS-CoV-2 infection of human bronchial epithelial cells (hBECs)[21] and human intestinal epithelial cells (hIECs)[22]. The hBECs data comprised transcripts from a mock infection, and cells at 1-, 2-, and 3-days post-infection (dpi) with SARS-CoV-2 isolate USA/USA-

WA1/2020[21]. The hIECs data was produced from colon- and ileum-derived organoids infected with SARS-CoV-2 isolate strain Germany/BavPat1/2020[22], and comprised transcripts from a mock infection, and infected cells at 12- and 24-hours post-infection (hpi). In both studies, gel bead-in emulsions (GEMs) were prepared from single-cell suspensions using the 10× Genomics Single Cell 3′ Library Kit NextGem V3.1 (10× Genomics, USA). hBECs libraries were sequenced on a NovaSeq600 system[21] and hIECs using a HiSeq4000 system[22]. In addition, a dataset involving infection of ileum organoids with human astrovirus 1 (HAstV1), and studied under similar conditions, was used as a control[73].

**Data processing**. Datasets were processed using custom Python scripts. Only sequences with a perfect match to the 10× genomics whitelist (3M-February-2018.txt.gz) and no ambiguous nucleotide calls in the UMI portion were used in the analysis. Selected sequences were mapped to a custom database comprising the complete set of reference human messenger RNAs available at NCBI, mitochondrial-encoded transcripts from the human reference mitochondrial genome (NC_012920), and SARS-CoV-2 genomes corresponding to the reference sequence Wuhan-Hu-1 (NC_045512) and isolate USA/USA-WA1/2020 (MW811435) and Germany/BavPat1/2020 (MZ558051) used for infection in both datasets (Supplementary Data 1). Read mapping was performed with MagicBLAST[74]. Only sequences that mapped unambiguously to a single gene with more than 95% coverage and a single UMI were included in this analysis. In contrast to the originally published analyses[21,22], we selected GEMs with respect to the total number of transcripts (UMIs) instead of the total number of transcribed genes. Empty GEMs and multiplets were removed from the datasets using a custom Python script that ordered GEMs with respect to the number of transcripts transformed using a $\log_{10}$ scale; then the local standard deviation of the ordered data was calculated using a window size of five datapoints. Upper and lower thresholds delimiting multiplets and void beads were determined using the 1.5×IQR rule (Supplementary Table 1 and Supplementary Fig. 1). Mapping information from the selected cells was then transformed into a matrix of counts which included genes detected in at least one hundred cells (Supplementary Data 1 and Supplementary Fig. 2). Furthermore, in contrast to the original analysis of these datasets, we did not discard cells with a high percentage of transcribed mitochondrial genes as overexpression of mitochondrial genes could be a legitimate response to SARS-CoV-2 infection[52,56].

**Estimation of gene frequencies**. The probability of detecting $n$ transcripts from a gene in a cell depends on its relative frequency and the total number of sampled transcripts. Therefore, as cDNA amplification of cellular transcripts represents only 10 to 20% of the total mRNA content[6,75,76], gene count data from scRNA-seq contains many missing values, or dropouts, that result from the Poisson sampling of low abundance transcripts in cells with few sequenced transcripts[9,76]. The probability of observing $n_{gi}$ read counts for a given gene $g \in \{1, 2,…, G\}$ across cells $i \in \{1, 2,…, I\}$ can be modeled using a Poisson distribution where the likelihood of observing $n_{gi}$ counts given a particular transcript abundance $\theta_{gi}$ is given by:

$$\mathcal{L}(\theta_{gi}) = p_{\theta_{gi}}(n_{gi}, N_i) = \frac{(\theta_{gi}N_i)^{n_{gi}} e^{-\theta N_i}}{n_{gi}!}, 0 \le \theta_{gi} \le 1 \quad (1)$$

$$\text{with } N_i = \sum_{g=1}^{G} n_{gi}$$

In this work, gene abundances were calculated as the weighted average of gene frequencies ($\theta$) with respect to the normalized likelihood function ($\hat{\mathcal{L}}$) and multiplied by a scaling factor of $10^4$ (transcripts per ten thousand or TPT):

$$TPT_{gi} = 10^4 \sum_{\theta=0}^{1} \theta_{gi} \hat{\mathcal{L}}(\theta_{gi}) \quad (2)$$

$$\text{with } \hat{\mathcal{L}}(\theta_{gi}) = \frac{\mathcal{L}(\theta_{gi})}{\sum_{i=1}^{I} \mathcal{L}(\theta_g)}.$$

An uncorrected normalized matrix was also calculated. Corrected and uncorrected frequency matrices are available in Supplementary Data 2.

**Volcano plot analysis**. Basal levels of expression for each gene were determined using the average expected TPT values from mock cells with at least one gene count per gene of interest. Infected cells, on the other hand, were binned with respect to viral loads using a $\log_2(TPT_{SARS-CoV-2} + 1)$ scale. Only cells with at least one count for SARS-CoV-2 and the target gene were used. $P$ values were calculated using a two-tailed Mann–Whitney $U$-test and corrected using the Benjamini–Hochberg method[77] using a significance threshold of 0.01. Differential expression analysis results are provided in Supplementary Data 3.

**Z-score profiles and analysis**. Normalized transcriptional responses were calculated using a z-score of the transcription levels observed at each $\log_2(TPT_{SARS-CoV-2} + 1)$ corrected by the levels and standard deviation observed in the uninfected cells. For each cell type, a mean response vector was calculated by averaging the normalized response of all genes. Pairwise comparisons between individual transcriptional profiles and the mean response vector were performed using a root-mean-square deviation (RMSD)

metric and the magnitude of the initial response ($\Delta_0$) was measured as the average z-score of the first five datapoints. A transcriptional response was considered an outlier when either the RMSD or $\Delta_0$ was outside the intervals defined by the 1.5×IQR rule.

**Gene ontology (GO) analysis**. Analysis of biological functions was performed using a custom database of high-quality GO terms of human proteins downloaded from the Uniprot database (UniProt Consortium, 2015; Supplementary Data 3). Quantification of the most abundant GO terms was performed using custom scripts that mapped selected genes to a dictionary of GO annotations. GO-enrichment analysis was performed by comparison of gene subsets to the list of mapped genes as background. $P$ values were calculated using Fisher's exact test and corrected Benjamini–Hochberg method (with a threshold significance of 0.01).

**Construction of correlation networks**. Undirected correlation networks were built using the normalized transcription profiles. A link between two genes was established when the two-sample Kolmogorov–Smirnov test indicated that both z-score distributions followed the same distribution ($P \le 0.01$) and the corrected Pearson correlation coefficient between expression profiles was $r \ge 0.9$. In both instances, $P$ values were corrected using the Benjamini–Hochberg method[77]. Only those genes which had at least 50 measurements in common were included in the analysis (Supplementary Data 5). Network properties were characterized with the Python networkX package[78] and visualized with Gephi[79].

**Statistics and reproducibility**. Details about the statistical analyses used in this study are described in the corresponding methods section. For volcano plots, the statistical significance was calculated using a two-tailed Mann–Whitney $U$-test. Transcriptional responses were classified as outliers when their RMSD or initial response ($\Delta_0$) was outside the intervals defined by the 1.5×IQR rule. For GO-enrichment analyses, $P$ values were calculated using Fisher's exact test. In the construction of the correlation network, correlations were measured using the two-sample Kolmogorov–Smirnov test on the Z-score distributions ($P \le 0.01$) and the corrected Pearson correlation coefficient between expression profiles ($r \ge 0.9$). In all these cases, $P$ values were corrected using the Benjamini–Hochberg method[77].

**Reporting summary**. Further information on research design is available in the Nature Portfolio Reporting Summary linked to this article.

## Data availability

The original datasets used in the analysis can be accessed at NCBI archived under BioProject accession codes PRJNA701930, PRJNA658984, and PRJNA720321. All data generated or analyzed during this study are included in this article and its supplementary information files. The Tables and processed datasets that support the findings of this study are available at the Zenodo repository (https://zenodo.org/) under the identifier (https://doi.org/10.5281/zenodo.7198900).

## Code availability

Codes that support the findings of this research are available at GitHub (https://github.com/paguties/SARS-CoV-2_scRNAseq.git).

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

## Acknowledgements

We thank Sergi Valverde, Salvador Durán-Nebreda, Irene Otero-Muras, the members of the I²SysBio-CRM UA DiscoVir, and our labmates for useful suggestions and productive discussions. This work was supported by European Commission – NextGenerationEU (Regulation EU 2020/2094) through CSIC's Global Health Platform (PTI + Salud Global) grants SGL2021-03-009 and SGL2021-03-052 to S.F.E.

## Author contributions

Conceptualization, Supervision, and Writing—reviewing and editing: P.A.G. and S.F.E. Data curation, investigation, Illustrations, Writing—original draft: P.A.G. Funding acquisition: S.F.E.

## Competing interests

The authors declare no competing interests.
