## [Peer Review File · Communications Biology]

Reviewers' comments:

Reviewer #1 (Remarks to the Author):

Gutierrez et al. re-analyzed scRNA-seq data from human bronchial epithelial cells, as well as intestinal organoids, to model transcriptional dynamics in response to SARS-CoV-2 infection. The study is clearly written and describes a well-thought-out model for temporal evolution of the transcriptional response to SARS-CoV-2 infection. The study findings, while not surprising or particularly dramatic, add an important layer of synthesis to the rapidly growing field of scRNA-seq data related to SARS-CoV-2. These types of integrated studies are much needed to distill lessons from the deluge of single cell multi-omics on SARS-CoV-2. Furthermore, the lessons learned from this manuscript in terms of the utility of z-score based analysis of temporal transcriptional responses may be widely applicable to other multi-timepoint or multi-condition scRNA-seq studies, where simple pairwise comparisons may bias the interpretation. The authors provide a detailed statistical framework that can be replicated in future studies. From my perspective, there are not major technical flaws to the study or manuscript, and I recommend acceptance with minor changes/comments, detailed below.

1. The abstract begins with a focus on scRNA-seq vs bulk RNAseq, which is a bit distracting and misleading. This paper is about understanding the transcriptional response to SARS-CoV-2 infection. The authors should consider re-focusing the abstract to reflect the structure of their introduction. This manuscript is not a systematic comparison of scRNA-seq vs bulk RNA-seq; rather, it is a pseudotime analysis of in vitro SARS-CoV-2 infection.
2. Can the authors clearly justify in the text when they selected these three datasets for analysis? Are these the only studies with temporal in vitro scRNA-seq of SARS-CoV-2 infection?
 - a. As a related point: the authors exclusively use data from in vitro systems, where temporal dynamics can be tightly controlled. Now that the authors have established clear temporal modules of transcription in response to SARS-CoV-2, is it possible to use these lessons to more thoughtfully interpret scRNA-seq data from in vivo human tissue with SARS-CoV-2 infection? Presumably, different cell types in vivo may exhibit different transcriptional dynamics, depending on virus tropism, among other factors?
3. Figure legends, generally speaking, are sparsely written and need additional detail in order to provide full explanation of the figures. Readers should be able to read the legends alone and understand the meaning of the figures. In particular, the description and interpretation of Fig 3D, 4C, and 5B is challenging.
4. Labeled genes in volcano plots (e.g. Fig 3C) are too small to read. Also, the text in GO plots (e.g. Fig 3B) is too small to read easily.
5. Custom python script used for GO analysis, but not found on GitHub.
6. The reviewer generally appreciated the discussion of the model put forth by the authors in the Discussion. While speculative, it was thoughtful and well-justified.
7. Minor grammatical issues:
 - a. Line 19 Abstract: should be "single cell" (singular)
 - b. "Bulked RNA-seq" should be changed to "Bulk" RNA-seq throughout manuscript
 - c. Lines 21-22: confusing verbiage, consider rephrasing
 - d. Line 186: "were" should be "where"
 - e. Line 187: repetitive use of the word "intermediate" is awkward, consider rephrasing
 - f. Line 193: authors abbreviate "differential gene expression" DEG instead of DGE. Should this be DGE?
 - g. Line 210: use of the word "regimes" here is awkward, consider rephrasing

Reviewer #2 (Remarks to the Author):

In their paper, Gutierrez and Elena use three previously published scRNA-seq datasets of cells infected with SARS-CoV-2 to determine the dynamics of the transcriptomic response to the viral infection. They identify a global wave-like transcriptional dynamics when using the viral load as a proxy for infection time. Using the average transcriptional profile over all genes as a reference along this pseudo-time of infection, they identify genes that appear to be outliers with respect to this average profile and discuss the functions of these genes, distinguishing between (1) ISG genes, (2) transcription factors and (3) genes encoding for RNA-binding proteins. Based on this analysis, the authors propose a model explaining the shape of the transcriptional response, involving the nsp1 gene as a central component involved primarily in repressing nuclear export of mRNAs (explaining the drop in the overall amount of cytoplasmic transcripts).

The analysis appears to be sound from a technical perspective, and the discussion opens an interesting hypothesis regarding the specific dynamics in the case of SARS-CoV-2. However, I also feel that some details are missing regarding technical aspects and that some controls need to be added.

- The authors have made a clear effort to optimize the computational analysis of the scRNA-seq datasets to make them comparable and suited for the analysis. In particular, they argue that the normalization procedure should be modified from total number of expressed genes to total numbers of UMIS. This is an interesting aspect, however I feel that the authors have not sufficiently justified and shown evidence for the possible bias they discuss in the last part (l. 438 - 445). They should present data showing the prevalence of viral transcripts over endogenous transcripts in the cell if they believe that this is a source of bias.

- The question remains of the specific transcriptional dynamics which is described is specific to SARS-CoV-2. I would strongly advise the authors to add a control dataset of a non-SARS-CoV-2 infection and verify if a similar or different dynamic is observed. Given that the authors have already analysed the Triana dataset, they could use another dataset from the same group describing an infection with astrovirus in the same tissue (Triana et al., MSB 2021).

- Given the use dynamic range of the transcriptional response, I am wondering if the average transcriptional profile discussed in figure 1c and used subsequently in all other analysis might be driven by a small set of highly expressed genes. Hence, rather than displaying an average global transcriptional profile, it actually represents the average expression profile of some highly expressed genes. I would therefore recommend to use median values instead of mean values (or any other approach to regularize the signal) to avoid this potential bias, which might impact the rest of the analysis.

- The different sections follow a similar scope, i.e. identifying genes that depart from the average expression profile. Apart from the concerns regarding a potential bias toward a few genes driving this average expression profile (see previous comment), the authors have not clearly explained what the rationale is behind identifying outlier genes. They identify enriched biological functions (Fig 2d), but why should these genes involved in these processes escape the general profile? I feel that this is not sufficiently justified in the beginning, nor sufficiently discussed in the Discussion. For example, the transcription factor YBX1 is identified as an outlier, but is not further mentioned in the discussion, and does not seem to contribute to the model discussed in the last section. So the identification of outlier genes appears as a nice exercise, but insufficiently motivated.

- some technicalities are not clear; for example, in figure 3a, the upper plots have z-scores as y-axis, while the lower row has logFC. It is unclear why plots use either of the 2 measures. In the same way, how are the plots 1a and 1c different? Here again, the two different measures are used.

- I do not understand the last sentence on page 21 (l. 367-369): how does the STRING network support different regulatory mechanisms? This is very obscure.

Minor comments

- Figures 3d and 3e do not seem to be referenced in the text
- Line 276: reference to Fig 3a seems wrong
- typos/missing word in line 402/499
- explain mRNA(c)/mRNA(n) in Figure 7

Responses to Reviewer 1 comments:

1. The abstract begins with a focus on scRNA-seq vs bulk RNAseq, which is a bit distracting and misleading. This paper is about understanding the transcriptional response to SARS-CoV-2 infection. The authors should consider re-focusing the abstract to reflect the structure of their introduction. This manuscript is not a systematic comparison of scRNA-seq vs bulk RNA-seq; rather, it is a pseudotime analysis of in vitro SARS-CoV-2 infection.

Answer: The reviewer is right. We have corrected the abstract to give a better description of the actual content of the paper. It now reads:

"Single-cell RNA sequencing (scRNA-seq) is currently one of the most powerful techniques available to study the transcriptional response of thousands of cells to an external perturbation. Here, we have performed a pseudotime analysis of SARS-CoV-2 infection using publicly available scRNA-seq data from human bronchial epithelial cells and colon and ileum organoids. Our results revealed that, for most genes, the transcriptional response to SARS-CoV-2 infection followed a non-linear pattern characterized by an initial and a final down-regulatory phases separated by an intermediate up-regulatory phase. A correlation analysis of transcriptional profiles suggests a common mechanism regulating the mRNA levels of most genes. Interestingly, genes encoded in the mitochondria or involved in translation exhibited distinct pseudotime profiles. To explain our results, we proposed a general model where nuclear export inhibition of nsp1-sensitive transcripts is sufficient to explain the transcriptional shutdown of SARS-CoV-2 infected cells."

2. Can the authors clearly justify in the text when they selected these three datasets for analysis? Are these the only studies with temporal in vitro scRNA-seq of SARS-CoV-2 infection?

Answer: The main goal of our study was to identify general mechanisms in the response to SARS-CoV-2 infection using virus accumulation as a proxy of time. To achieve this, we selected our dataset using the following criteria:

- 1) scRNA-seq data where the SARS-CoV-2 response was studied at different macroscopic time points that included at least, one uninfected control.
- 2) Datasets generated under similar experimental conditions. In our case, all the data was obtained the 10X genomics platforms.
- 3) To include studies involving different cell-types to be able to discriminate between general and cell-specific responses.
- 4) To include data from different groups to remove lab-dependent biases.
- 5) To use cultured cells to avoid the complexities derived from the immune response.

At the time datasets from Ravindra et al. (2021) and Triana et al. (2021) were the only ones that satisfied these criteria. In addition, to follow the suggestion of one of the reviewers we have now also included a control dataset, involving infection of ileum organoids with human astrovirus 1 (HAstV1), and studied under similar conditions (Triana et al., 2021b). The conclusion is that the pseudo-time profiles for SARS-CoV-2 and HAstV1 are different, supporting the notion that cellular responses to a great extent are virus-specific (or driven by the virus).

a. As a related point: the authors exclusively use data from in vitro systems, where temporal dynamics can be tightly controlled. Now that the authors have established clear temporal modules of transcription in response to SARS-CoV-2, is it possible to use these lessons to more thoughtfully interpret scRNA-seq data from in vivo human tissue with SARS-CoV-2 infection? Presumably, different cell types in vivo may exhibit different transcriptional dynamics, depending on virus tropism, among other factors?

Answer: This is a very good point. Our study describes the effect of SARS-CoV-2 on infected cells in the absence of an immune response. Performing a similar study to this one will add an extra layer of complexity

that we believe will require additional methods to deconvolute the effects of the immune system. We are currently investigating these effects using a network approach, and this will likely be the topic of a follow-up study.

3. Figure legends, generally speaking, are sparsely written and need additional detail in order to provide full explanation of the figures. Readers should be able to read the legends alone and understand the meaning of the figures. In particular, the description and interpretation of Fig 3D, 4C, and 5B is challenging.

Answer: Figure legends have been corrected following the reviewer's suggestion. More details have been provided to allow the independent interpretation of each figure. Changes are highlighted in the corrected text. We have also redrawn some of the figures to make them more readable and removed some panels that did not add much to the discussion.

4. Labeled genes in volcano plots (e.g. Fig 3C) are too small to read. Also, the text in GO plots (e.g. Fig 3B) is too small to read easily.

Answer: As detailed in the previous response, we have increased the size of the labels to improve readability.

5. Custom python script used for GO analysis, but not found on GitHub.

Answer: We apologize for not having it available. We have uploaded the missing script.

6. The reviewer generally appreciated the discussion of the model put forth by the authors in the Discussion. While speculative, it was thoughtful and well-justified.

Answer: Thank you very much for your appreciation of our toy model.

7. Minor grammatical issues:

a. Line 19 Abstract: should be "single cell" (singular)

Answer: Corrected.

b. "Bulked RNA-seq" should be changed to "Bulk" RNA-seq throughout manuscript

Answer: Corrected.

c. Lines 21-22: confusing verbiage, consider rephrasing

Answer: The corresponding sentence was removed from the corrected abstract.

d. Line 186: "were" should be "where"

Answer: Corrected

e. Line 187: repetitive use of the word "intermediate" is awkward, consider rephrasing

Answer: The corresponding sentence now reads: "An intermediate phase takes place at medium viral loads and is characterized by an increase in expression with maximum fold changes of between 1.4 and 4.1 in ileum".

f. Line 193: authors abbreviate “differential gene expression” DEG instead of DGE. Should this be DGE?

Answer: It should be DGE as highlighted by the reviewer. All instances have been corrected to DGE.

g. Line 210: use of the word “regimes” here is awkward, consider rephrasing

Answer: the corresponding sentence now reads: "After this transformation, intervals of up- or down-regulation correspond to positive and negative values, respectively".

Responses to Reviewer 2 comments:

- The authors have made a clear effort to optimize the computational analysis of the scRNA-seq datasets to make them comparable and suited for the analysis. In particular, they argue that the normalization procedure should be modified from total number of expressed genes to total numbers of UMIS. This is an interesting aspect, however I feel that the authors have not sufficiently justified and shown evidence for the possible bias they discuss in the last part (l. 438 - 445). They should present data showing the prevalence of viral transcripts over endogenous transcripts in the cell if they believe that this is a source of bias.

Answer: In our preliminary analyses, we noticed that selecting cells with respect to the number of detected genes could decrease the number in infected cells at high viral accumulations when the virus becomes the dominant transcript. This makes sense as the SARS-CoV-2 could be detected at about 9464.5, 7915.2, and 9291.8 transcripts per ten thousand in the hBECs, colon, and ileum organoids, respectively. For example, in their analysis, Triana et al. (2021) removed cells with less than 1,500 detected genes. If we had used that filtering criterium, the number of infected cells used in the analysis would decrease from 30,033 to 16,054 in hBECs, from 8,132 to 4,282 in colon, and 7,553 to 4,746 in Ileum. As shown in the attached figure, this effect is more pronounced at high SARS-CoV-2 counts. To highlight this point, we have included this figure as panel D in Supplementary Figure 2.

- The question remains of the specific transcriptional dynamics which is described is specific to SARS-CoV-2. I would strongly advise the authors to add a control dataset of a non-SARS-CoV-2 infection and verify if a similar or different dynamic is observed. Given that the authors have already analysed the Triana dataset, they could use another dataset from the same group describing an infection with astrovirus in the same tissue (Triana et al., MSB 2021).

Answer: We have followed the reviewer's suggestion and analyzed 36 datasets from Triana et al. (2021b) SRR14163089-24 corresponding to ileum organoids comprising mock infections and infections at 4 h and 16 h post-infection with human astrovirus 1 (HAstV1). The corresponding count and frequency matrices have been provided as Supplementary Data 6. Z-score plots for each gene were also calculated and available in the same link.

We analyzed a total of 5,895 cells comprising 11,936 genes. As can be seen from the following figure, the average response upon HAstV1 seems to be very different from the observed profile in SARS-CoV-2 in the same cell type (Ileum cells). Moreover, the genes that exhibited correlated profiles in SARS-CoV-2 infected cells, do not seem to be very different from the average response (blue). A strong wave-like behavior is not observed in this case. These figures have been added as Supplementary Figure 3.

We have also added the following sentences in the Results section:

To verify the specificity of the SARS-CoV-2 response, we performed a similar analysis on ileum cells infected HAstV1 (Supplementary Fig. 3) which revealed a different profile characterized by a rapid drop in transcription levels followed by a steady increase at high virus accumulations. A strong wave-like behavior was not observed in this case

and

Finally, an analysis of the transcriptional profiles in cells infected with HAstV1 did not reveal a distinct correlated responses in genes from modules 1, 2, and 3 (Supplementary Figure 3), suggesting that transcriptional clusters are probably specific to SARS-CoV-2.

And the following text in the Discussion section:

This behavior seems to be a feature of SARS-CoV-2 infection as a similar analysis performed on ileum cells infected with HAstV1 did not reveal a correlated response for most genes, and did not exhibit an intermediate up regulatory phase (Supplementary Fig. 3, Supplementary Data 6)

- Given the use dynamic range of the transcriptional response, I am wondering if the average transcriptional profile discussed in figure 1c and used subsequently in all other analysis might be driven by a small set of highly expressed genes. Hence, rather than displaying an average global transcriptional profile, it actually represents the average expression profile of some highly expressed genes. I would therefore recommend to use median values instead of mean values (or any other approach to regularize the signal) to avoid this potential bias, which might impact the rest of the analysis.

Answer: The reviewer's concern is justified. However, a bias in the average response vector resulting from high expressing genes is unlikely in our analysis for the following reasons:

- i. We normalized each gene expression profile using z-scores. As this is a relative scale, extreme values are not expected to bias the average response profile.
- ii. If highly expressed genes had a dominant effect in the average response profile, it would not be expected that about 90% could be fitted to this profile.
- iii. An analysis of expression levels in the mock treatments shows that outliers have higher than normal expression levels, supporting the notion that the average response is not the result from biased highly expressed genes. This is illustrated in the following figure, where the module that deviated most

from the average profile (modules 2 and 3) have expression levels above average. This proves that the normalization worked well to remove biases from highly expressed genes.

To further clarify this point, we have made added the following clarifications in the text:

In the Results section: *"To do this, expression values at each \log_2 interval were transformed into z scores with respect to the mean and standard deviations of uninfected cells. This correction removes biases in the mean response patterns that might result from highly-expressed genes."*

In caption to Figure 1: *"c Average transcriptional profile for each cell type. Each plot was calculated from individual gene profiles scaled using z-scores with respect to the transcript levels observed in uninfected cells."*

- The different sections follow a similar scope, i.e. identifying genes that depart from the average expression profile. Apart from the concerns regarding a potential bias toward a few genes driving this average expression profile (see previous comment), the authors have not clearly explained what the rationale is behind identifying outlier genes.

They identify enriched biological functions (Fig 2d), but why should these genes involved in these processes escape the general profile? I feel that this is not sufficiently justified in the beginning, nor sufficiently discussed in the Discussion.

Answer: At the beginning of this investigation, we wanted to explain the non-linear response to SARS-CoV-2 and hypothesized that the observed patterns were the result of a common mechanism involving genes involved in the interferon response, transcription factors or RNA binding proteins. To clarify this point we added the following sentence:

"As the differential transcriptional response might be the result of general regulatory mechanisms induced or inhibited by SARS-CoV-2, we decided to interrogate the data to identify common responses involving interferon signaling, transcription factors, and RNA binding proteins."

At the end of the ISG section we added:

"However, it does not look like there is any obvious connecting between these genes and the global transcriptional profiles observed upon SARS-CoV-2 infection."

For example, the transcription factor YBX1 is identified as an outlier, but is not further mentioned in the discussion, and does not seem to contribute to the model discussed in the last section.

Answer: When we analyzed transcription factors, our hope was to identify an outlier that was general enough to help us explain the down-regulatory and/or up-regulatory profiles observed in the global transcriptional response. We did not discuss YBX1 further as it provide us with little explanatory power. We added the followed to the text:

"These results suggest that transcriptional response of most TFs is non-specific; however, exceptions probably include TFs expressed after the initial transcriptional shutdown induced after the activation of cell-death pathways at late SARS-CoV-2 infection stages. Again, a connection between JUN, JUND or YBX1 and the global transcriptional response is not clear."

- some technicalities are not clear; for example, in figure 3a, the upper plots have z-scores as y-axis, while the lower row has logFC. It is unclear why plots use either of the 2 measures. In the same way, how are the plots 1a and 1c different? Here again, the two different measures are used.

Answer: The reviewer was right about the second plots being confusing. They were intended to illustrate the enrichment in GO terms associated with the antiviral response. A similar plot was also included in one of the panels of Figure 4 in the submitted paper. After reviewing the manuscript, we realized these panels did not add much information and have been removed from the manuscript.

- I do not understand the last sentence on page 21 (l. 367-369): how does the STRING network support different regulatory mechanisms? This is very obscure.

Answer: We have edited the text to make our point clearer.

"Finally, a search in the STRING database, revealed that most abnormal responding RBPs are part of highly connected physical network which suggests a distinct response to SARS-CoV-2 infection of genes implicated in translation (Fig. 5e)."

As in the caption of Figure 5e:

"A STRING database search of differentially regulated RBPs revealed a highly connected functional network suggesting that transcripts involved in translation are regulated differently with respect to most genes."

Minor comments

- Figures 3d and 3e do not seem to be referenced in the text

Answer: The corresponding panel have been referenced in the text. Due to figure changes, panels 3d and 3e are now 3b and 3c. The corresponding text reads:

"normal mucosa of esophagus-specific gene 1 protein (C15orf48), and interferon alpha-inducible protein 27 (IFI27) (Fig. 3c)."

"Examples of ISGs exhibiting different response are shown in figure (3d)."

- Line 276: reference to Fig 3a seems wrong

Answer: The reviewer is right, the reference was wrong and was corrected.

- typos/missing word in line 402/499

Answer: The missing words have been added as shown:

line 402: To investigate the presence of of potential sequence motifs

line 499 "a mechanism used to modulate several critical"

- explain mRNA(c)/mRNA(n) in Figure 7.

Answer: There was a mistake in panel b. It has been corrected as it explicitly indicates mRNA (nucleus) and mRNA (cytoplasm) which were labelled as mRNA(c) and mRNA(n) in the original figure:

b

c

REVIEWERS' COMMENTS:

Reviewer #1 (Remarks to the Author):

Thank you for the detailed response. All concerns have been addressed

Reviewer #2 (Remarks to the Author):

The authors have made a significant effort in answering the comments made in the first review and performed the requested additional analysis. I have no further comment at this point, apart maybe from the fact that z-score scaling does not prevent from outlier effect in general, as both the mean and standard deviation used in the z-score are highly susceptible to outliers, which might impact the z-score. Using median and IQR in the z-score computation does however prevent this effect. However, the authors provided additional arguments for the lack of outlier bias, which I found convincing. I do recommend acceptance of this article.

Responses to Reviewer #1

R1.Q1. Thank you for the detailed response. All concerns have been addressed.

Thank you so much for all your constructive comments. They've been very useful to improve the final version of the manuscript.

Responses to Reviewer #2

R2.Q1. The authors have made a significant effort in answering the comments made in the first review and performed the requested additional analysis. I have no further comment at this point, apart maybe from the fact that z-score scaling does not prevent from outlier effect in general, as both the mean and standard deviation used in the z-score are highly susceptible to outliers, which might impact the z-score. Using median and IQR in the z-score computation does however prevent this effect. However, the authors provided additional arguments for the lack of outlier bias, which I found convincing.

Thank you for all your very constructive comments and suggestions. We believe the manuscript has been largely improved by incorporating them.

I do recommend acceptance of this article.

Thank you so much.